# Oracle-MoE: Locality-preserving Routing in the Oracle Space for Memory-constrained Large Language Model Inference

Jixian Zhou [* 1]  Fang Dong [* 1]  Ruijun Huang [* 1 2]  Hengjie Cao [1]  Mengyi Chen [1]  Yifeng Yang [1]  Anrui Chen [1]
Mingzhi Dong [3]  Yujiang Wang [4]  Dongsheng Li [5]  David A. Clifton [4]  Qin Lv [6]  Rui Zhu [7]  Chun Zhang [8]
Fan Yang [9]  Tun Lu [1]  Ning Gu [1]  Li Shang [1]

.

## Abstract

Mixture-of-Experts (MoE) is widely adopted to deploy Large Language Models (LLMs) on edge devices with limited memory budgets. Although MoE is, in theory, an inborn memory-friendly architecture requiring only a few activated experts to reside in the memory for inference, current MoE architectures cannot effectively fulfill this advantage and will yield intolerable inference latencies of LLMs on memory-constrained devices. Our investigation pinpoints the essential cause as the remarkable temporal inconsistencies of inter-token expert activations, which generate overly frequent expert swapping demands dominating the latencies. To this end, we propose a novel MoE architecture, Oracle-MoE, to fulfill the real on-device potential of MoE-based LLMs. Oracle-MoE route tokens in a highly compact space suggested by attention scores, termed the *oracle space*, to effectively maintain the semantic locality across consecutive tokens to reduce expert activation variations, eliminating massive swapping demands. Theoretical analysis proves that Oracle-MoE is bound to provide routing decisions with better semantic locality and, there-fore, better expert activation consistencies. Experiments on the pretrained GPT-2 architectures of different sizes (200M, 350M, 790M, and 2B) and downstream tasks demonstrate that without com-promising task performance, our Oracle-MoE has achieved state-of-the-art inference speeds across varying memory budgets, revealing its substantial potential for LLM deployments in industry.

## 1. Introduction

The unprecedented prevalence of Large Language Models (LLMs) calls for their sink from remote servers to local edge devices for better accessibility, privacy protections, and personalizations. Mixture-of-Experts (MoE) (Fedus et al., 2022; Cai et al., 2024) is the most widely adopted technique for the on-device deployment of LLMs. A well-established advantage of MoE is that it could effectively scale up LLMs' capacities without increasing the inference computational complexity, which is vital for their executions on edge devices of less powerful computing hardware. However, we would like to underline a less-mentioned but crucial property of MoE: it is, in theory, a memory-friendly architecture that could naturally satisfy the memory con-straint of edge devices. Only a few activated experts are needed to reside in the device memory for a feedforward MoE inference, and thus, the minimum memory occupa-tions of MoE-based LLMs can be significantly less than their dense counterparts, which indicates the edge-friendly nature of MoE architectures.

In practice, the fulfilling of MoE's memory-saving advan-tage, however, is still far from satisfying. To date, deploy-ing MoE-based LLMs on memory-constrained devices will commonly introduce intolerable inference latencies that sub-stantially impair the industrial values of LLMs. Those la-tencies are predominantly caused by the overheads from frequently loading activated experts into the memory and cleaning old ones during successive inferences of tokens. As shown in Figure 1(a) (*Top*), the memory-constrained in-ference of MoE-based LLMs can result in latencies approxi-

*Equal contribution [1]College of Computer Science and Arti-ficial Intelligence, Fudan University, Shanghai, China [2]Greater Bay Area National Center of Technology Innovation, Research Institute of Tsinghua University in Shenzhen, Shenzhen, China [3]University of Bath, Bath, UK [4]Oxford Suzhou Centre for Ad-vanced Research, Suzhou, China [5]Microsoft Research Asia, Shang-hai, China [6]Department of Computer Science, University of Col-orado Boulder, Colorado, USA [7]Bayes Business School, City St George's, University of London, London, UK [8]Research Institute of Tsinghua University in Shenzhen, Shenzhen, China [9]School of Microelectronics, Fudan University, Shanghai, China. Corre-spondence to: Li Shang <lishang@fudan.edu.cn>, Yujiang Wang <yujiang.wang@oscar.ox.ac.uk>.

*Proceedings of the $42^{nd}$ International Conference on Machine Learning*, Vancouver, Canada. PMLR 267, 2025. Copyright 2025 by the author(s).

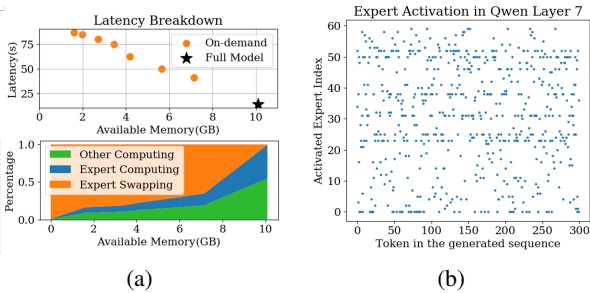

*Figure 1.* a). *Top*: compared with a full model residing in the memory, memory-constrained inference of LLMs suffers from severe latency issues; *Bottom*: 50%-85% latencies are incurred by the I/O overheads from expert swapping. An edge device with 8GiB GPU memory is adopted. b). Visualization of 60 expert activations of layer 7 in Qwen (Yang et al., 2024) across a sequence of 300 tokens.

mately 15 to 30 times higher than the same model entirely residing in the memory. 50%-85% of those latencies, as portrayed in Figure 1(a) (*Bottom*), were incurred by the I/O overheads from expert swappings. Despite several prior attempts (Kong et al., 2024; Yi et al., 2023; Huang et al., 2023b; Zhang et al., 2024; Rajbhandari et al., 2022) to optimize those latencies, they followed an engineering-based perspective to develop comparatively naive and empirical strategies, such as quantization (Zhu et al., 2023), expert pruning (Cheng et al., 2023; Rajbhandari et al., 2022), retaining frequently-used experts (Kong et al., 2024; Yi et al., 2023; Huang et al., 2023b; Artetxe et al., 2022), which lack systematic, in-depth investigations.

This work thoroughly examines the underlying mechanism behind the high latencies of memory-constrained MoE-based LLM inference. Figure 1(b) depicts the inter-token activations of 60 experts of one layer across a sequence of 300 tokens, which exhibit a highly spare and temporally inconsistent pattern. Such a pattern will inevitably create a plethora of expert-swapping demands to be resolved, and it is challenging to develop an effective empirical strategy due to the complexity, sparsity, and inconsistency of activations. The essential cause for this pattern can arguably be attributed to that current MoE routings are predominantly influenced by token embeddings, which tend to be dominated by token-specific information with high inter-token variations.

To this end, we devise Oracle-MoE, a novel MoE architecture to generate routing decisions that naturally activate identical experts across consecutive tokes to reduce inter-token variations, eliminating massive expert-swapping needs and accelerating memory-constrained inference. Oracle-MoE is bridged on a characteristic of edge scenarios such that the linguistic meaning between consecutively generated tokens

is typically consistent, and successive queries from a user usually share similar knowledge grounds. In other words, a token sequence represents notable *semantic locality* across nearby tokens. Inspired by this observation, Oracle-MoE conducts routing in a highly compact space, termed the *oracle space* to assign tokens instead of the usual token embeddings.

Particularly, our analysis revealed that tokens with higher mutual attention scores share similar high-level semantics, which we call semantic groups. To efficiently extract high-level semantics, we use average token embeddings in each group as semantic group embeddings and denote the set of these embeddings as *oracle space*. Empirical evidence and theoretical analysis demonstrate that this concise calculation of oracle space captures semantic locality and routing in the oracle space could preserve semantic locality better when selecting experts, considerably decreasing the swapping frequencies. On GPT-2-based networks (Radford et al., 2019) of different capacities (200M, 350M, 790M, and 2.06B) and the downstream tasks including classification, QA and summarization, our Oracle-MoE has attained state-of-the-art inference speeds than those baseline strategies across varying memory budges without comprising the performance.

## 2. Analysis

In this section, we begin with defining the latency optimization problem in semantic space with introducing the **C**onsecutive **S**emantic **D**ifference (**CSD**). Given that consecutive tokens show semantic localities, we aim to find a low-variance semantic embedding for each token. We first model the token embeddings as the combination of high-level semantics and token-identity semantics. Then, we use attention scores to discover high-level semantics similarity and obtain semantic groups, and construct the **Oracle Space** with semantic group embeddings. Finally, we show that oracle-space-based routing yields significantly lower CSD compared to token-level routing, both theoretically and empirically.

### 2.1. Latency Problem Formalization

Deploying Mixture of Expert (MoE) models on edge devices faces the challenge of expert swapping latency. Meanwhile, we do not want latency optimization to degrade the model's performance. So, we introduced a constraint that the latency optimization must ensure that the measurement of performance $M$ that the user is interested in is better than a threshold $\gamma$:

$$\min L_{total} \quad \textbf{s.t. } M > \gamma$$

Our experiments in Section 5 will show that latency optimization does not necessarily lead to performance degradation. For simplicity, we omit the subject to item in the

following of this section. The total latency $L_{\text{total}}$ can be written as:

$$L_{\text{total}} = \sum_{t=1}^{T} L_{\text{compute}}(t) + L_{\text{swap}}(e_t, e_{t-1}),$$

where $L_{\text{compute}}(\mathbf{t}_t)$ denotes computation latency for token embedding $\mathbf{t}_t$, and $e_t$ denotes the set of activated expert indices for token $\mathbf{t}_t$. $L_{\text{swap}}(e_t, e_{t-1})$ captures swapping latency caused by loading experts that are activated by $\mathbf{t}_t$ but not activated by $\mathbf{t}_{t-1}$. Specifically:

$$L_{\text{swap}}(e_t, e_{t-1}) = l_{\text{swap}} \cdot |e_t \setminus e_{t-1}|,$$

where $|e_t \setminus e_{t-1}|$ is a set minus operation. In practice, MoE-based LLMs on edge devices generate continuous semantics, the large variance in the token-level MoE routing mechanism is essentially due to the high inter-token semantic variance, resulting in frequent changes in routing results. We define **C**onsecutive **S**emantic **D**ifference (**CSD**) to measure the variation in expert selection over consecutive tokens:

$$\text{CSD} = \sum_{t=2}^{T} \Delta e_t,$$

Given fixed hardware and swapping algorithms, $L_{compute}$ and $l_{swap}$ are fixed, thus **CSD** determines $L_{\text{total}}$. The final optimization goal is:

$$\min \sum_{t=2}^{T} \Delta e_t.$$

**Remark:** Setting $\Delta e_t = 0$ leads to a dense model that would trivially minimize CSD, which is not considered in our work as it negates the fundamental advantage of MoE models—leveraging multiple experts for specialized processing.

In existing token-level MoE models, experts are usually selected through a linear gate $W_g \in \mathbb{R}^{N \times d}$ (Cai et al., 2024), which projects the token embedding $\mathbf{t}_t \in \mathbb{R}^d$ into expert scores $g(\mathbf{t}_t) = W_g \mathbf{t}_t \in \mathbb{R}^N$. The experts are then chosen as:

$$e_t = \text{Top-k}_{i \in \{1, \dots N\}}(\text{softmax}(W_g \mathbf{t}_t)_i)$$

where Top-k denotes the indices of top k largest elements. Inspired by previous work (Xue et al., 2024), current token-level MoEs tend to dispatch on token-identity semantics. So, we approximate the change in Top-k expert selection between consecutive tokens by the difference in their transformed embeddings:

$$\text{CSD}_{token} = \sum_{t=2}^{T} \Delta e_t \approx \sum_{t=2}^{T} C(W_g, k) \|\mathbf{t}_t - \mathbf{t}_{t-1}\|$$

where $C(W_g, k)$ is a constant dependent on $W_g, k$.

However, since there is a large variance between token embeddings, as is shown in Figure 2 left, the CSD is hard to optimize for existing token-level MoE. Xue et al. (2024) also mentioned that token-level MoEs tend to dispatch tokens according to token ID. But the linguistic meaning of consecutive user interactions on edge devices is notably similar, which we term as *semantic locality*. **_This inspires us to minimize swapping latency by extracting high-level semantics and designing a new gating mechanism_**.

## 2.2. Semantic Groups and Oracle Space

**Token Embeddings** In token-level MoEs, tokens in the same sequence do share some similar high-level semantics, with their high-level semantics' locality and tendency to cluster logically, as shown in Figure 2. But their embeddings are still influenced primarily by the token-identity features, leading us to explore semantic spaces. These tokens simultaneously contain two types of semantic information.

**Definition 1** (Token Embedding). *For each token embedding* $\mathbf{t}_i$,

$$\mathbf{t}_i = \mathbf{s}_i + \mathbf{u}_i,$$

*where* $\mathbf{s}_i$ *represents shared high-level semantic information between consecutive tokens and* $\mathbf{u}_i$ *represents unique token-identity information.*

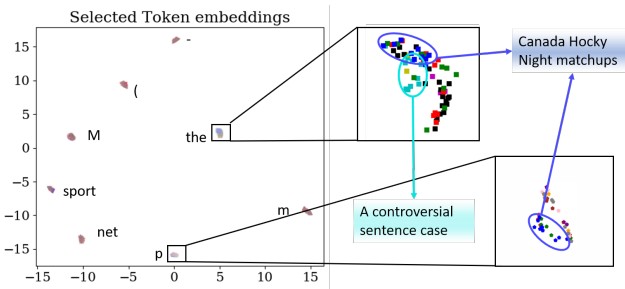

*Figure 2.* UMAP visualization of embedding space in existing token-level MoE models. Left: Tokens tend to cluster according to token-identity semantics. Right: Tokens from the same sequence are colored the same. They share similar semantics and stay closer to each other in each token cluster.

**Semantic Groups** Studies ((Vaswani et al., 2017; Kovaleva et al., 2019; Clark et al., 2019)) indicate that attention captures high-level semantic correlations between tokens. Therefore, we intuitively claim that the mapping of the Q/K matrices and the calculation of the attention scores will group consecutive tokens with similar high-level semantics together through significant attention score distribution differences. Our analysis in Appendix A.1 demonstrates that under our definition of token embedding and analysis of the Q/K matrix, a high attention score between tokens indicates

that they share similar high-level semantics. So, we propose to model this through a causal graph perspective, where semantic groups emerge from connectivity in the attention score matrix.

Consider a directed acyclic graph $G_{\text{global}} = (V_{\text{global}}, E_{\text{global}})$, where $V_{\text{global}}$ contains all tokens and $E_{\text{global}}$ consists of edges $\mathbf{t}_i \to \mathbf{t}_j$ weighted by $a_{ij}$ from the lower-triangular attention score matrix $A_{\text{global}} = [a_{ij}]$ (i.e., $a_{ij}$ exists only for $i > j$). We define semantic groups as maximally connected components that only tokens with an attention score larger than a predefined threshold $\epsilon$ are considered connected:

**Definition 2** (Semantic Group). *A subset containing tokens $S = \{\mathbf{t}_{k_1}, ..., \mathbf{t}_{k_m}\}$ (indices $k_1 < ... < k_m$) is called a semantic group if:*

$$\begin{cases} \forall i, j \in \{k_1, \ldots, k_m\}, \ (i > j) \implies a_{ij} > \epsilon \\ \textit{No proper superset of } S \textit{ satisfies the condition above} \end{cases}$$

This can be regarded as a reformulation of the Minimum Clique Cover Problem (Gavril, 1972) for DAGs. The definition leverages the block structure of attention score matrices as is shown in Figure 3, which is also observed in previous works((Liu et al., 2024)). So, although the Minimum Clique Cover is NP-hard, we claim that it can be computationally tractable on attention score matrix via polynomial-time greedy algorithms (Farjas, 2018). We first initialize each token as a singleton group, then for the token $x_i$ from left to right, we find the maximal $j < i$ with $a_{ij} > \epsilon$ and merge $x_i$ into $x_j$'s group if $\forall x_k$ in the group, $a_{ik} > \epsilon$.

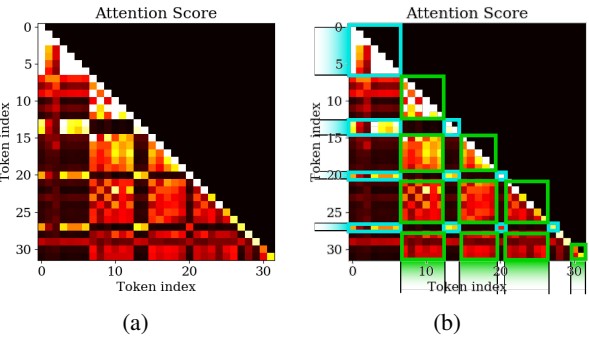

*Figure 3.* Visualization of attention score matrix. There are two semantic groups where tokens in each group show high attention scores with each other.

**Discussion:** Previous studies (Kamath et al., 2019) on representation space analysis have shown that semantically similar samples exhibit higher similarity in their embeddings compared to semantically dissimilar ones, which is also widely validated in experiments with general-purpose large models. We corroborate this observation and further identify a more fine-grained similarity pattern: token representations encapsulate both high-level semantics and token identity

semantics. Among tokens with the same identity, the embeddings of those that share the same high-level semantic meaning tend to be more similar. This pattern is consistently observed in various models, including widely used large models like DeepSeek-16B-2.8B (Dai et al., 2024) and Qwen1.5-MoE-A2.7B (Team, 2024), which are illustrated in Figure 2, Figure 10 and Figure 11 in Appendix B.2. Theoretical insights into how attention mechanisms compute correlations between tokens using the inner product of query (Q) and key (K) vectors are also supported by existing studies (Raffel et al., 2020; Vig & Belinkov, 2019). The computation of attention scores involves first assessing token correlations through inner products of query (Q) and key (K) vectors, followed by normalization of these correlations via softmax, and finally allocating contextual information through value (V) vectors weighted by the normalized scores. Among which, the Q-K inner product effectively captures token similarity and reflects high-level semantic alignment, as visualized in Appendix B.2.

**Oracle Space** Following sentence meta-embedding techniques (Poerner et al., 2019; Takahashi & Bollegala, 2022), we compute semantic group embeddings as the average token embeddings in it. Formally:

**Definition 3** (Semantic Group Embedding). *For a Semantic Group $S_i$, its semantic group embedding $\mathbf{z}_{S_i}$ is defined as:*

$$\mathbf{z}_{S_i} = \frac{1}{|S_i|} \sum_{\mathbf{t}_j \in S_i} \mathbf{t}_j$$

As proven in (Xu et al., 2018; Soltanolkotabi et al., 2013) and demonstrated in Appendix A.2, this aggregation reduces token-identity noise while preserving essential high-level semantics. Thus, we can efficiently extract various high-level semantic information from the embedding space using semantic group embeddings. We collect semantic group embeddings from different data and name the space consisting of these embeddings as **Oracle Space**, efficiently describing various high-level semantics.

In inference task, new tokens arrive sequentially over time. To model the evolution of semantic groups and derive token embeddings based on these groups, we propose the semantic embedding of each token as its semantic group embedding. Given a token $\mathbf{t}_t$ at time step $t$, let $S(t)$ denote the semantic group corresponding to $\mathbf{t}_t$. We use the embedding of $S(t)$ as the token's semantic embedding:

$$\mathbf{z}_{\mathbf{t}_t} = \mathbf{z}_{S(t)} = \frac{1}{|S(t)|} \sum_{\mathbf{t} \in S(t)} \mathbf{t},$$

The $S(t)$ includes all tokens from previous time steps $t' < t$ such that:

$$\forall \mathbf{t}_{t'} \in S(t), \quad a_{tt'} > \epsilon,$$

where $a_{tt'}$ is the attention score between tokens $\mathbf{t}_t$ and $\mathbf{t}_{t'}$. When the model generates consecutive tokens, it retains the KV cache of previous tokens, so that $a_{tt'}$ can be obtained by adding a new row to $A_{\text{global}}$.

This provides a way to compute semantic groups with a streaming input, which is the case for auto-regressive generation. As is shown in Figure 4, in an auto-regressive generation process, the token's semantic embedding varies smoothly and slowly in the oracle space, preserving high-level semantic information.

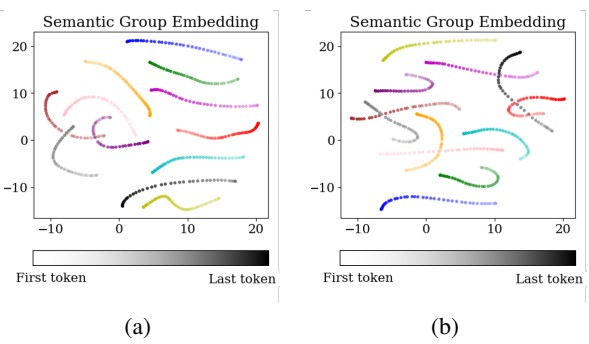

First token      Last token       First token      Last token

(a)                   (b)

*Figure 4.* UMAP visualization of sampled semantic group embeddings (token's semantic embeddings) in different model layers. Each color represents a sequence(user interaction). As token generation goes on, the embeddings based on semantic groups vary slowly and smoothly.

## 2.3. Oracle-Space-Based Routing Yields Lower CSD

Based on the above analysis, routing tokens with similar semantic embedding in the oracle space to the same expert is highly likely to yield a low expert variation. Since semantic group embeddings evolve locally in the oracle space (Figure 4), we employ clustering algorithms to allocate tokens that belong to the same cluster in the oracle space to the same expert, or:

$$e_t = \arg\min_k \|\mathbf{z}_{\mathbf{t}_t} - c_k\| = \arg\min_k \|\mathbf{z}_{S(t)} - c_k\| \quad (1)$$

where $c_k$ denotes the cluster center in oracle space, which will be detailed in Section 3. So, the change in expert assignments can be approximated by the change of semantic group of $\mathbf{z}_{\mathbf{t}_t}$. We provide a detailed analysis of this trend in the AppendixA.3, and the following form:

**Definition 4** (CSD for oracle-space-based routing). *Let $S(t-1)$ and $S(t)$ denote the semantic groups of token $\mathbf{t}_t, \mathbf{t}_{t-1}$ at consecutive time steps. Without loss of generality, based on semantic variations on semantic groups, we have CSD of oracle-space-based routing:*

$$CSD_{oracle} = \sum_{t=2}^{T} \Delta e_{t,oracle} \approx \sum_{t=2}^{T} \left\| \mathbf{z}_{S(t)} - \mathbf{z}_{S(t-1)} \right\|$$

Next, we highlight the low-variation gains of oracle-space-based routing compared to token-level routing, which reduces unnecessary routing changes for semantically similar tokens. This leads to the following theorem:

**Theorem 1.** *With a high probability, $\forall t \in \mathbb{N}, \mathbf{t}_t, \mathbf{t}_{t-1}, a_{tt-1} > \epsilon$ with a sufficiently high dimension of token identity information:*

$$C(W_g, k)\|\mathbf{t}_t - \mathbf{t}_{t-1}\| > \left\| \mathbf{z}_{S(t)} - \mathbf{z}_{S(t-1)} \right\|$$

Through the CSD estimation framework established in Section 2.1, this induces the relationship:

$$\text{CSD}_{\text{token}} > \text{CSD}_{\text{oracle}},$$

This indicates that oracle-space-based routing provides lower variation by reducing routing changes for semantically similar tokens. The detailed proof is provided in the Appendix A.4.

## 3. Oracle-MoE

Based on the above analysis, we propose Oracle-MoE, replacing the token-level MoE routing mechanism with a better semantic locality in the oracle space and expert activation consistencies for edge-side devices to generate text scenarios.

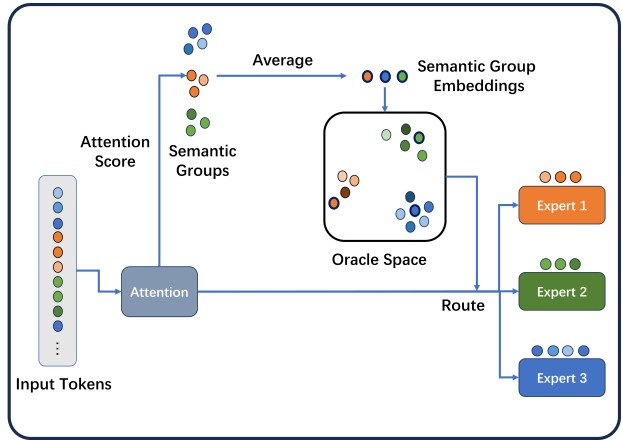

*Figure 5.* Overview of an Oracle-MoE layer. Residual connections are omitted.

**Oracle Space Initialization** We obtain the initial oracle space after a short warm-up training phase of the token-level MoE model. After the warm-up stage, we randomly sample $N$ data, and get the semantic group embeddings of each data as is mentioned in Section 2, Definition 2, and 3. These semantic group embeddings form an initial oracle space. Routing in the oracle space does not necessitate complete information; only distinguishing high-level semantics is required. Therefore, to improve computational

efficiency, we adopt SVD to reduce these embeddings to lower dimensions (Schmidt, 2020).

**Oracle-MoE Routing at Pretraining**  Since the semantic group embedding varies only locally within a small region in the oracle space as tokens are generated, to ensure tokens in the same semantic group to be dispatched to the same expert, we run K-means (Jain, 2008) in the oracle space to get $k$ oracle clusters. The parameter $k$ here is equal to the expert number of the original token-level MoE, and Section 5 will show that our method yields persistently good results with $k$ ranging from 4 to 32. In the following training process, we replace the routing of the original token-level MoE with the routing mechanism shown in Figure 5. For each new incoming data, we first divide it into semantic groups according to the attention scores and get their semantic group embeddings in the oracle space with the same SVD transform matrix computed in the previous stage. Then we calculate which oracle cluster each semantic group belongs to as Equation 1, and dispatch tokens in the semantic group to the corresponding expert.

**Oracle-MoE Routing at Inference**  The inference stage can be divided into prefill and decode stages. The prefill stage is the same as the training stage routing. In the decode stage, Oracle-MoE first decides which semantic group the coming token belongs to, and updates the semantic group embedding with the coming token. Since KV cache is a widely adopted strategy (Yuan et al., 2024), this does not introduce memory overhead. Then we dispatch it to the expert corresponding to the oracle cluster of its semantic group. In our experiments, there are often fewer than 5 semantic groups in an input session within a length of 1024, and semantic groups from the same session are likely to belong to the same oracle cluster. So, the oracle space routing preserves the semantic locality of input tokens and yields a low expert variation, contributing to low expert swapping latency.

## 4. Related Works

**Mixture of Experts**  The Mixture of Experts (MoE) is a fundamental model in machine learning (Jacobs et al., 1991; Jordan & Jacobs, 1994) and an instance of the conditional computation framework where different experts are responsible for different inputs. To increase the model's capacity to deal with complex data, (Eigen et al., 2013) extended the MoE structure to deep neural networks and proposed a deep MoE model composed of multiple layers of routers and experts. (Shazeer et al., 2017) simplified the MoE layer by making the output of the gating function sparse for each example, which greatly improves the training stability and reduces the computational cost. Since then, the MoE layer with different base neural network structures (Shazeer et al.,

2017; Dauphin et al., 2017; Vaswani et al., 2017) has been proposed and achieved great success in a variety of tasks. MoE has been widely explored to improve the training efficiency of Large Language Models(LLMs), with various routing strategies like (i) letting tokens select the top-k experts (Lepikhin et al., 2021; Fedus et al., 2022; Zuo et al., 2022; Chi et al., 2022; Dai et al., 2022; Chen et al., 2023), (ii) letting experts select the top-k tokens (Zhou et al., 2022), to (iii) globally decide expert assignment (Lewis et al., 2021; Clark et al., 2022).

**Memory-constrained Inference Solutions for Mixture of Experts**  Huang et al. propose swapping experts from GPU memory to CPU memory to reduce memory usage, though this approach introduces significant latency overhead (Huang et al., 2023a). SE-MoE (Shen et al., 2022) employs Ring Memory offloading to minimize GPU memory consumption. EdgeMoE (Yi et al., 2023), the on-device inference engine tailored for MoE-based LLMs, reduces expert I/O overhead through expert-wise bitwidth adaptation and in-memory expert management, achieving low-latency inference while maintaining task performance. While Swap-MoE (Kong et al., 2024), by dynamically managing a small set of Virtual Experts based on activation locality and hardware profiling, reduces memory consumption and latency, but sacrifices accuracy. These works do not take into account the characteristics of semantic space, especially the cluster characteristics of high-level semantics and in-depth investigations.

## 5. Experiments

### 5.1. Settings

**Hardware Platform**  Since mainstream mobile phones(like Android and Apple ) and NPU manufacturers (e.g., Apple, Hisilicon, Qualcomm, Samsung) do not provide commercial APIs for low-level GPU memory operations, we adopt NVIDIA Jetson Xavier NX as our experimental platform. The NVIDIA Jetson Xavier NX is equipped with a 384-core NVIDIA Volta architecture GPU with 8 GiB of GPU memory and an estimated 21 TOPS AI computing power.

**Models**  We mainly compare Oracle-MoE with Switch Transformer (Fedus et al., 2022), a representative token-level MoE architecture. Experts are loaded on demand under our experiment settings. We use models containing $m$ MoE layers with $n$ experts each and with a total parameter of $p$, denoted as $n * m(p)$. In our experiments, we use models of 2*4(192M), 4*8(295M), 8*16(729M) and 9*24(2.06B). Detailed model configurations are in Appendix B.

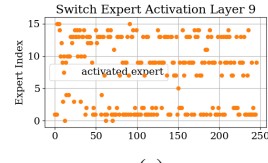
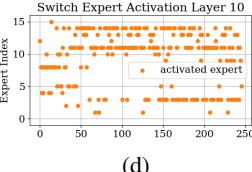

*Figure 6.* Expert activation results of (8*16)729M models. In Switch Transformer, almost every 2 consecutive tokens activate different experts, and nearly all experts are activated, demanding frequent expert swapping. Whereas Oracle-MoE requires only a few expert swappings as the token generation continues.

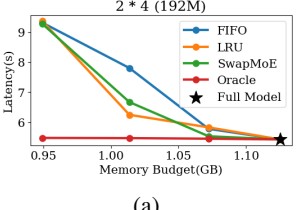
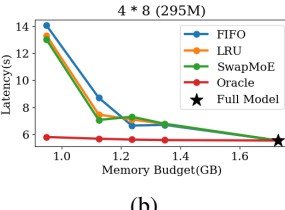
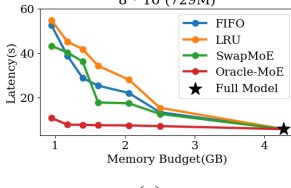
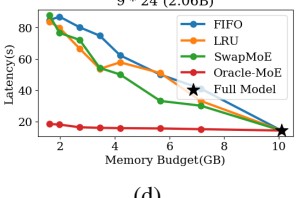

*Figure 7.* Memory-Latency curve of models of different sizes, where $n * m(p)$ denotes a model consisting of $n$ MoE Layers with $m$ experts each, and in total $p$ parameters. Full model refers to a model with the same number of activated parameters.

**Swapping Strategies** As introduced in Section 1, swapping strategies assign a certain priority to each expert so that lower-priority experts can be swapped out first. We evaluate both our model and the Switch Transformer with First-In-First-Out (FIFO), Least Recently Used (LRU), and the strategy in SwapMoE (Kong et al., 2024). FIFO swaps out the expert that is loaded first, and LRU swaps out the experts that have not been used for the longest time. In SwapMoE(Kong et al., 2024), experts are weighted by their frequency, magnitude, and input tokens. However, since our model eliminates the requirement of expert swapping, different strategies don't make a big difference in our model. So, we report the average of 3 strategies on our model.

**Data & Workload** Models are pretrained on Openweb-Text (Komatsuzaki, 2019), which is one of the pretraining datasets of GPT2 (Radford et al., 2019). We primarily use downstream tasks of 3 types: question answering, classification, and summarization. For QA tasks, we adopt Trivia QA (Joshi et al., 2017). For classification, we adopt GLUE (Wang et al., 2019), MAG (Sinha et al., 2015) and Sci-Cite (Beltagy et al., 2019). We also use XSum (Narayan et al., 2018) for summarization tasks. In experiments, we always kept the batch size equal to 1, which is the real situation when running on edge devices like mobile phones, processing one user request at a time.

**Metrics** We adopt mainly 3 evaluation metrics. a) **Expert Activation**, to evaluate the variation in expert activation of different models. b) **Memory-latency** curve measures the average time the model takes to process a single data point

for a given memory size. A larger memory provides models with redundancy to store temporarily unused experts and mitigate the penalty of expert activation variation. c) **First token Latency** measures the time before the first token is generated after the users provide the input. This is also an important metric for user experience.

### 5.2. Results

**Expert Activation** Figure 6 illustrates the expert activation of two models. During consecutive auto-regressive generation passes, our method shows a lower expert activation variation, where expert swapping is only triggered after hundreds of tokens are generated. In Switch Transformer, expert activation changes frequently.

**Memory-Latency** Figure 7 illustrates the memory-latency curve for methods of different sizes. The result of our model is reported as the average of different strategies. As is shown in the figure, while with a small-sized model, the latency is acceptable, the case becomes worse rapidly as the model size gets larger. For the 8*16(729M) model, even though only 1 expert is allowed for each layer (which is about only 25% of the full-size memory), our method introduces only 3s additional latency compared with the full-size inference. Whereas Switch Transformer with FIFO, LRU, or Swap-MoE load-on-demand strategy introduces inevitable latency, increasing latency by up to 2000% compared to a full-size memory inference. When the memory budget increases by up to 50% of the full-size model, the latency of the Switch Transformer with different strategies is still unacceptably high, whereas our model does not introduce latency.

| Size | Model | TrivialQA(F1) | GLUE(Acc.) | MAG(Acc.) | Sci-Cite(Acc.) | Xsum(Rouge-1) | Avg. |
|------|-------|---------------|------------|-----------|----------------|---------------|------|
| 195M | Switch | 27.00 | 62.25 | 20.00 | 30.83 | 13.55 | 30.73 |
|      | Ours  | 26.72 | 62.86 | 18.33 | 32.50 | 13.36 | 30.75 |
| 295M | Switch | 30.10 | 64.76 | 22.67 | 31.25 | 14.62 | 32.68 |
|      | Ours  | 30.31 | 64.66 | 22.17 | 33.75 | 15.53 | 33.28 |
| 729M | Switch | 35.08 | 68.33 | 25.29 | 35.00 | 15.62 | 35.86 |
|      | Ours  | 35.56 | 68.00 | 25.50 | 36.67 | 16.05 | 36.35 |
| 2.06B | Switch | 46.06 | 77.75 | 31.33 | 46.75 | 16.77 | 43.73 |
|      | Ours  | 46.96 | 78.00 | 30.67 | 47.50 | 17.35 | 44.09 |

*Table 1.* The zero-shot performance of models of different sizes on different tasks. Metrics reported depend on specific tasks. Our method does not pose a drawback to model performance on downstream tasks.

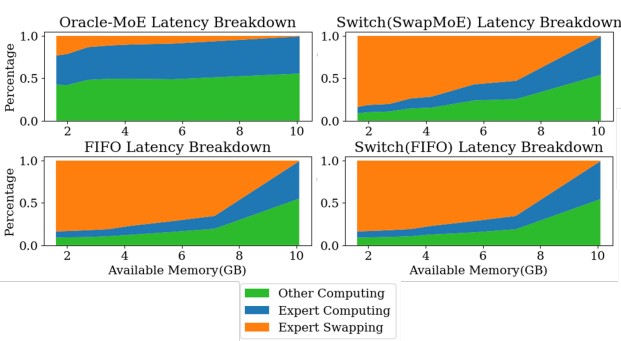

| Model(Strategy) | First token latency(s) |
|-----------------|------------------------|
| Switch(FIFO) | 22.395 |
| Switch(LRU) | 23.428 |
| Switch(SwapMoE) | 12.767 |
| Oracle-MoE | **4.910** |

*Table 2.* First token latency of 765M models on different architectures(strategies) under 50% of full-size memory. The memory budget only influences SwapMoE since it uses offline statistics to determine which expert to load first.

*Figure 8.* Latency composition of our proposed Oracle-MoE and Switch Transformer equipped with different swapping strategies. Our model reduces the percentage of expert swapping latency and thus reduces the overall latency.

### 5.3. Overhead Analysis

**Performance on Downstream Tasks**   Although designed for edge-oriented scenarios, our model does not sacrifice performance on downstream tasks for edge-deploy inference latency. As is shown in Table 1, our proposed semantic group gating method shows a similar task performance, in some tasks even surpasses, the widely accepted token-level gating MoE models. We believe this is attributable to our proposed semantic group-level routing strategy. This setup allows each expert to focus on a subset of high-level semantics rather than requiring every expert to learn all possible high-level semantics present in their target tokens, thereby reducing redundancy among experts.

The latency composition breakdown in Figure 8 gives a detailed visualization of the above results. It can be observed that even with the most limited memory budget, the latency introduced by expert swapping in our model only contributes to 50% of the overall latency. In token-level routing, such as the switch transformer, expert swapping contributes to more than 99% of latency.

**First token latency**   Our model activates fewer experts for a single input, so that only 1 or 2 experts are needed for the prefilling stage. So, our model requires only one-time expert loading during the prefilling stage. However, existing token-level MoE methods still need to swap experts during the prefilling stage, leading to worse first token latency. Among the three expert swapping strategies, FIFO does not help at the prefilling stage, and the LRU strategy needs a warm-up stage to decide on frequently used tokens. SwapMoE, however, uses off-line statistical information to decide the loaded expert at the beginning of inference, thus resulting in a lower first token latency than baselines, but still not as good as ours.

**Training Stage Overhead**   Our approach differs from existing token-level MoE in that it includes a one-time cluster analysis after the warm-up phase and cluster routing in each forward pass. In our experiments, with a sample size of 8192, the wall clock time for clustering analysis per layer is approximately 4 min, which is negligible compared with tens of hours of pretraining. For routing in each pass, token-level MoE is equivalent to performing a matrix multiplication. It requires 1e-4 seconds, whereas our low-dimensional cluster Euclidean distance computation requires three matrix multiplications and a square root operation. Thanks to the low-dimensional semantic space, the final wall-clock time

of our routing is 2.5e-4 seconds, which is also negligible compared to the single forward-backward pass taking 3.5 seconds.

### 5.4. Expert Prediction-based Optimization

The Oracle-MoE has significantly diminished the necessity for expert swapping, thereby fundamentally reducing latency. However, the load-on-demand strategy still inherently suffers from another limitation: it cannot decide which expert to load for a layer before that layer is reached. Consequently, we propose to predict deep layers' expert activation at shallower layers, enabling inferring current tokens and loading experts synchronously. Specifically, we use the embeddings of the first layer to predict the expert activation in the following layers. Experimental results show that Oracle-MoE reaches an expert prediction accuracy of 85% to 95%, whereas Switch-Transformer-like token-level routing structure only has an accuracy of 40% to 60%. Employing this can further reduce the expert loading latency of Oracle-MoE by 10% to 15%. The underlying reasons for this expert predictability will be left for future investigation.

## 6. Conclusion

This paper introduces Oracle-MoE, an innovative MoE architecture designed to optimize edge deployment by addressing latency issues inherent in token-level routing. By incorporating an Oracle-Space-Based routing mechanism, Oracle-MoE reduces expert activation variation and minimizes memory swapping overhead. Extensive experiments validate that Oracle-MoE achieves lower latency without introducing any drawback on downstream tasks or pretraining computation, which is vital for the on-device deployments of LLMs. Future work could extend this approach to other conditional computation frameworks and explore its applicability across diverse hardware environments.

## Impact Statement

This work aims to deliver insights to the Machine Learning community and will not lead to any direct societal consequences. While it is associated with LLMs that may output misleading or harmful content, such issues are outside the scope of this work and will not be particularly specified here.

## Acknowledgements

Yujiang Wang was supported by a Basic Research Program of Jiangsu (BK20240414) and a Leadership Talent Program (Science and Education) of Suzhou Industrial Park (KJQ2024204).

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

# A. Proof of Analysis

## A.1. Property of semantic element

In this section, we delve into the detailed analysis of the approximate attention mechanism, specifically focusing on the impact of the inner product between query ($W_Q$) and key ($W_K$) matrices.

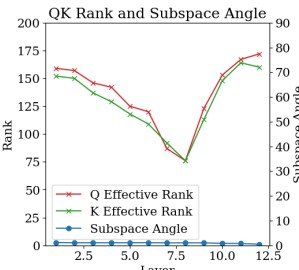

*Figure 9.* Effective rank of matrices $W_Q$ and $W_K$ in different layers' self-attention and their subspaces angle. The space determined by $W_Q$ and $W_K$ is very similar.

The attention score computation can be viewed as a mapping from two token representations to a scalar value:

$$\mathcal{A}: \mathcal{H} \times \mathcal{H} \to \mathbb{R},$$

In Figure 9, $W_Q$ and $W_K$ exhibit similar ranks and minimal angles between their subspaces. Combined with the empirical success of shared QK matrices in transformers (e.g., Kowsher et al. (2024)), we propose that $W_Q$ and $W_K$ can be decomposed into a shared projection followed by subspace-specific transformations. We therefore propose:

$$W_Q = M_Q P, \quad W_K = M_K P,$$

where $P: \mathcal{H} \to \mathcal{H}_\mathcal{S}$ is a shared projection matrix mapping tokens to a common subspace $\mathcal{H}_\mathcal{S}$ of dimension $r$ (with $r \leq \dim(\mathcal{H})$), and $M_Q, M_K : \mathcal{H}_\mathcal{S} \to \mathcal{H}_\mathcal{S}$ are full-rank linear transformations within $\mathcal{H}_\mathcal{S}$.

Given this understanding:

**Assumption 1** (Semantic Subspace). *The token representation space $\mathcal{H}$ can be decomposed into a low-dimensional semantic subspace $\mathcal{H}_s$ and its orthogonal complement $\mathcal{H}_S^\perp$:*

$$\mathcal{H} = \mathcal{H}_\mathcal{S} \oplus \mathcal{H}_S^\perp.$$

Followed by the analysis and the experiment results in Section 2.2, although we recognize that $W_Q \neq W_K$, meaning $M_Q \neq M_K$, under our analysis, where both $M_Q$ and $M_K$ are different full-rank linear mappings from $\mathcal{H}_\mathcal{S}$ to $\mathcal{H}_\mathcal{S}$, we can derive a relatively symmetric bound for the inner products involving these mappings. Specifically, for any non-zero, unit vectors $X, Y \in \mathcal{H}_\mathcal{S}$ , we have:

$$|\langle M_Q X, M_K Y\rangle| \leq \|M\| \cdot |\langle X, Y\rangle|,$$
$$|\langle M_Q Y, M_K X\rangle| \leq \|M\| \cdot |\langle Y, X\rangle|.$$

where $\|M\|$ denotes the operator norm (spectral norm) of matrix $M_Q^T M_K$ This bound reflects how the transformations $M_Q$ and $M_K$ affect the original inner product $\langle X, Y\rangle$.

Furthermore, although the above discussion is based on an approximate attention mechanism and focuses on the properties of token components in the $\mathcal{H}_S$ space, Figure 2 shows that tokens with the same identity (i.e., semantically similar tokens) tend to cluster together in low-dimensional representations that preserve relative distances. But this clustering behavior also suggests that tokens with the same high-level semantic meaning have small relative distances in the semantic component space. Therefore, for tokens within the same subgraph or semantic context, we make the following assumption:

**Assumption 2.** *For tokens $\mathbf{t_i}$ in the same subgraph, or Semantic Group $S_j$, their components in semantic space $\mathcal{H}_\mathcal{S}$ $\mathbf{s_i}$, there exist $r$, s.t. $\mathbf{s}_i \in B_{\mathcal{H}_\mathcal{S}}(c_i, r)$*

**Assumption 3** (Uniform Distribution in Semantic Space). *Tokens in the same semantic group $s_j$ are uniformly distributed in the semantic space with an expected value at the center:*

$$\mathbb{E}[\mathbf{s}_i] = \mathbf{c}_j, \quad \forall i \in s_j.$$

## A.2. The advantage of mean embedding of subgraph

Given the definition of semantic groups, we now make an assumption about the distribution of token identity information $\mathbf{u_i}$, which refers to the unique characteristics within each subgraph, distinguishing one token from another beyond their shared high-level semantics:

**Assumption 4** (Normal Distribution in Token Identity Space). *Token identical information follows a normal distribution within each subgraph $S_j$:*

$$\mathbf{u}_i \sim \mathcal{N}(\mu_j, \Sigma_j), \quad \forall i \in S_j.$$

Then we have The variance of the semantic representation of token $\mathbf{t_i}$ is defined as:

$$\mathrm{Var}(\mathbf{z}_{\mathbf{t}_t}) = \mathrm{Var}\left(\frac{1}{|S(t)|} \sum_{\mathbf{t} \in S(t)} \mathbf{t}\right)$$

Let the neighborhood size be $n = |S(t)|$, then:

$$\mathbf{z}_{S(t)} = \frac{1}{n} \sum_{i \in S(t)} \mathbf{t}_i = \frac{1}{n} \sum_{i \in S(t)} (\mathbf{s}_i + \mathbf{u}_i) \quad (2)$$

It can be decomposed into two parts:

$$\mathbf{z}_{S(t)} = \underbrace{\frac{1}{n} \sum_{i \in S(t)} \mathbf{s}_i}_{\text{Mean of semantic part}} + \underbrace{\frac{1}{n} \sum_{i \in S(t)} \mathbf{u}_i}_{\text{Mean of identity part}} \quad (3)$$

Variance decomposition:

$$\text{Var}(\mathbf{z}_{S(t)}) = \text{Var}\left(\frac{1}{n}\sum \mathbf{s}_i\right) + \text{Var}\left(\frac{1}{n}\sum \mathbf{u}_i\right)$$
$$+ 2\text{Cov}\left(\frac{1}{n}\sum \mathbf{s}_i, \frac{1}{n}\sum \mathbf{u}_i\right) \quad (4)$$

Then we have the analysis of each component:

1. **Covariance term**: Given the orthogonal subspace decomposition $\mathcal{H}_s \perp \mathcal{H}_s^{\perp}$, the semantic part $\mathbf{s}_i$ and the identity part $\mathbf{u}_i$ are independent, hence the covariance is zero:

$$\text{Cov}\left(\frac{1}{n}\sum \mathbf{s}_i, \frac{1}{n}\sum \mathbf{u}_i\right) = 0 \quad (5)$$

2. **Variance of the semantic part**: According to Assumption 3 (uniform distribution), within the same subgraph, $\mathbf{s}_i$ are independently and identically distributed with $\mathbb{E}[\mathbf{s}_i] = \mathbf{c}_j$. Let $\text{Var}(\mathbf{s}_i) = \Sigma_s$, then:

$$\text{Var}\left(\frac{1}{n}\sum \mathbf{s}_i\right) = \frac{1}{n^2}\sum_{i=1}^{n}\text{Var}(\mathbf{s}_i) = \frac{\Sigma_s}{n} \quad (6)$$

3. **Variance of the identity part**: According to Assumption 4 (normal distribution), within the same subgraph, $\mathbf{u}_i \sim \mathcal{N}(\mu_j, \Sigma_j)$ and they are independent, then:

$$\text{Var}\left(\frac{1}{n}\sum \mathbf{u}_i\right) = \frac{1}{n^2}\sum_{i=1}^{n}\text{Var}(\mathbf{u}_i) = \frac{\Sigma_j}{n} \quad (7)$$

thus we have

$$\text{Var}(\mathbf{z}_{S(t)}) = \frac{\Sigma_s + \Sigma_j}{n} < \text{Var}(\mathbf{t}_t) \quad (8)$$

### A.3. Approximation Analysis of Oracle CSD

The expert assignment change $\Delta e_t$ is defined as the symmetric difference between consecutive expert sets:

$$\Delta e_t = |e_t \setminus e_{t-1}|$$

For simplicity, assume each token activates a single expert, so $\Delta e_t = \mathbb{I}(e_t \neq e_{t-1})$ (0 or 1).

When $\mathbf{z}_{S(t)}$ and $\mathbf{z}_{S(t-1)}$ reside in the same cluster, $\Delta e_t = 0$. When they lie in different clusters, $\Delta e_t = 1$. Let $\mathcal{B}$ denote cluster boundaries in the oracle space. The probability of crossing $\mathcal{B}$ between $t-1$ and $t$ increases with $\|\mathbf{z}_{S(t)} - \mathbf{z}_{S(t-1)}\|$.

For small displacements, we approximate the discrete boundary-crossing event by the continuous embedding displacement:

$$\Delta e_t \approx \|\mathbf{z}_{S(t)} - \mathbf{z}_{S(t-1)}\| \cdot \frac{\text{Cluster density at } \mathcal{B}}{\text{Cluster volume}}$$

Under uniform cluster assumptions, the density-to-volume ratio simplifies to a constant, yielding:

$$\sum_{t=2}^{T}\Delta e_t \approx \sum_{t=2}^{T}\|\mathbf{z}_{S(t)} - \mathbf{z}_{S(t-1)}\|$$

### A.4. Proof of Theorem 1

First, we have this lemma:

**Lemma 5** (Norm Comparison with Additive Threshold).
*For $n < m$, $\|Y\| + \mathbf{m} < K\|Z\|$ holds with probability approaching 1 as $d \to \infty$, where $\mathbf{m} > 0$ and $K > 0$ are fixed constants.*

*Proof.* Let $\{x_i\}_{i=1}^{m}$ be i.i.d. $d$-dimensional Gaussian vectors with:

- $\mathbb{E}[x_i] = \mu \in \mathbb{R}^d$
- $\text{Cov}(x_i) = \sigma^2 I_d$, where $I_d$ is the $d \times d$ identity matrix.

Let $S = \{k_1, \dots, k_n\}$ be a uniformly random subset of $\{1, \dots, m\}$ (without replacement) with $n < m$. Define:

$$Y = \frac{1}{m}\sum_{i=1}^{m}x_i - \frac{1}{n}\sum_{j=1}^{n}x_{k_j}, \quad Z = x_1 - x_2.$$

**For $Y$:**

$$\mathbb{E}[Y] = 0,$$
$$\text{Cov}(Y) = \sigma^2\left(\frac{1}{n} - \frac{1}{m}\right)I_d,$$
$$\mathbb{E}[\|Y\|^2] = d\sigma^2\left(\frac{1}{n} - \frac{1}{m}\right).$$

**For $Z$:**

$$\mathbb{E}[Z] = 0,$$
$$\text{Cov}(Z) = 2\sigma^2 I_d,$$
$$\mathbb{E}[\|Z\|^2] = \text{tr}(\text{Cov}(Z)) = 2d\sigma^2.$$

For $n < m$:

$$\frac{1}{n} - \frac{1}{m} < 2$$
$$\implies \mathbb{E}[\|Y\|^2] = d\sigma^2\left(\frac{1}{n} - \frac{1}{m}\right)$$
$$< 2d\sigma^2 = \mathbb{E}[\|Z\|^2].$$

Define the modified gap $W_K = K^2\|Z\|^2 - \|Y\|^2$. We analyze:

$$P(\|Y\| + \mathbf{m} < K\|Z\|) = P\Big(\|Y\|^2 + 2\mathbf{m}\|Y\|$$
$$+ \mathbf{m}^2 < K^2\|Z\|^2\Big).$$

*Table 3.* Hyperparameters of Models

| Hyperparameters | 195M MoE | 295M MoE | 729M MoE | 2.06B MoE |
|---|---|---|---|---|
| Attention heads | 12 | 12 | 12 | 16 |
| Transformer layers | 12 | 12 | 12 | 24 |
| MoE layers | 2 | 4 | 8 | 9 |
| Expert Number | 4 | 8 | 16 | 32 |
| Activated Expert Number | 1 | 1 | 1 | 1 |
| Hidden dimension size | 768 | 768 | 768 | 1024 |
| Dropout | 0.1 | 0.1 | 0.1 | 0.1 |
| Attention dropout | 0.1 | 0.1 | 0.1 | 0.1 |
| Sequence length | 256 | 256 | 512 | 1024 |
| Batch size | 320 | 320 | 160 | 80 |
| Learning rate decay | Cosine | Cosine | Cosine | Cosine |
| Maximum Learning rate | 4e-4 | 4e-4 | 2e-4 | 1e-4 |

| Activation inconsistency | DeepSeek | Qwen | Switch | Oracle |
|---|---|---|---|---|
| 1st 1/4 layers avg | 80.84 | 81.56 | 69.20 | **6.03** |
| 2nd 1/4 layers avg | 65.35 | 71.04 | 64.87 | **4.82** |
| 3rd 1/4 layers avg | 70.68 | 75.37 | 53.36 | **4.20** |
| 4th 1/4 layers avg | 76.61 | 77.16 | 75.44 | **5.11** |

*Table 4.* Activation inconsistency comparison across layers.

Using Chebyshev's inequality for $W_K$:

$$\mathbb{E}[W_K] = K^2 \mathbb{E}[\|Z\|^2] - \mathbb{E}[\|Y\|^2] = d\sigma^2 \left( 2K^2 - \frac{1}{n} + \frac{1}{m} \right),$$

$$\mathrm{Var}(W_K) = K^4 \mathrm{Var}(\|Z\|^2) + \mathrm{Var}(\|Y\|^2) \quad \text{(independence)},$$

$$= 8K^4 d\sigma^4 + 2d\sigma^4 \left( \frac{1}{n} - \frac{1}{m} \right)^2.$$

Set $\epsilon_K = \mathbb{E}[W_K] - 2m\sqrt{\mathbb{E}[\|Y\|^2]} - m^2$. Substituting $\mathbb{E}[\|Y\|^2]$:

$$\epsilon_K = d\sigma^2 \left( 2K^2 - \frac{1}{n} + \frac{1}{m} \right) - 2m\sqrt{d\sigma^2 \left( \frac{1}{n} - \frac{1}{m} \right)} - m^2.$$

Applying Chebyshev's inequality:

$$P\left( W_K \geq \epsilon_K \right) \geq 1 - \frac{\mathrm{Var}(W_K)}{\epsilon_K^2}.$$

Thus:

$$P\left( \|Y\| + m < K\|Z\| \right) \geq$$

$$1 - \frac{8K^4 d\sigma^4 + 2d\sigma^4 \left( \frac{1}{n} - \frac{1}{m} \right)^2}{\left[ d\sigma^2 \left( 2K^2 - \frac{1}{n} + \frac{1}{m} \right) - 2m\sqrt{d\sigma^2 \left( \frac{1}{n} - \frac{1}{m} \right)} - m^2 \right]^2}.$$

The dominant terms scale as:

$$\mathrm{Var}(W_K) = O(d), \quad \epsilon_K = \Theta(d).$$

Therefore:

$$\frac{\mathrm{Var}(W_K)}{\epsilon_K^2} = O\left( \frac{1}{d} \right) \to 0$$

$$\implies \lim_{d \to \infty} P\left( \|Y\| + \mathbf{m} < K\|Z\| \right) = 1.$$

**Note:** we also give the analysis of the impact of parameter constrains: when $n$ approaches $m$:

$$\frac{1}{n} - \frac{1}{m} \approx 0 \implies \begin{cases} \mathbb{E}[W] \approx 2K^2 d\sigma^2 - m^2, \\ \mathrm{Var}(W) \approx 8K^4 d\sigma^4 + 4m^2. \end{cases}$$

The probability bound becomes:

$$P(W > 0) \geq 1 - \frac{8K^4 d\sigma^4 + 4m^2}{(2K^2 d\sigma^2 - m^2)^2}.$$

For large $d$, the dominant terms yield:

$$P(W > 0) \geq 1 - \frac{8K^4 d\sigma^4}{4K^4 d^2 \sigma^4} = 1 - \frac{2}{d}.$$

So if we make an extreme assumption about the left and right components in semantic space, that is

$$(\mathbf{t}_t - \mathbf{t}_{t-1})|_{\mathcal{H}_S} = 0, (\mathbf{z}_{S(t)} - \mathbf{z}_{S(t-1)})|_{\mathcal{H}_S} = 2r$$

$$\|\mathbf{z}_{S(t)} - \mathbf{z}_{S(t-1)}\| = \left\| \left(\mathbf{z}_{S(t)} - \mathbf{z}_{S(t-1)}\right)\Big|_{\mathcal{H}_S}^{\perp} \right\|$$

$$+ \left\| \left(\mathbf{z}_{S(t)} - \mathbf{z}_{S(t-1)}\right)\Big|_{\mathcal{H}_S} \right\|$$

$$= \left\| \left(\mathbf{z}_{S(t)} - \mathbf{z}_{S(t-1)}\right)\Big|_{\mathcal{H}_S}^{\perp} + 2r \right\|.$$

$$\|(\mathbf{t}_t - \mathbf{t}_{t-1})|_{\mathcal{H}^{\perp}{}_S}\| = \|(\mathbf{t}_t - \mathbf{t}_{t-1})\|$$

Since $(\mathbf{t}_t - \mathbf{t}_{t-1})|_{\mathcal{H}^{\perp}{}_S}, (\mathbf{z}_{S(t)} - \mathbf{z}_{S(t-1)})|_{\mathcal{H}_S}^{\perp}$ follows Lemma5, so let $2r$ , $C(W_g, k)$ to be $\mathbf{m}, K$ in Lemma 5, with a existing $d$ mentioned in lemma 5,

$$C(W_g, k)\|(\mathbf{t}_t - \mathbf{t}_{t-1})|_{\mathcal{H}^{\perp}{}_S}\| > \left\|\mathbf{z}_{S(t)} - \mathbf{z}_{S(t-1)})|_{\mathcal{H}_S}^{\perp} + 2r\right\|$$

Thus we have

$$C(W_g, k)\|(\mathbf{t}_t - \mathbf{t}_{t-1})\| > \left\|\mathbf{z}_{S(t)} - \mathbf{z}_{S(t-1)})\right\|$$

$\square$

## B. Experiments

### B.1. Pretraining Hyperparameters

Table 3 lists the hyperparameters used in our experiments.

### B.2. Activation Inconsistency and Semantic Locality in Existing MoEs

We propose temporal activation inconsistency, defined as the average number of inconsistent expert activations per 100 consecutive tokens per expert. Results over the entire dataset and across different models and layers are listed in Table 4. Existing MoEs show strong temporal activation inconsistency within all layers, while Oracle-MoE reduces this.

Experiments with DeepSeekMoE-16B and Qwen1.5-MoE-A2.7B on real chat datasets(Wizard-of-Wikipedia and Synthetic-Persona-Chat) are shown in Figure 10 and Figure 11. Semantic locality appears across different models/layers/samples. Semantic groups can still be distinguished based on attention score and obtained by our method, as shown in Figure 12 to Figure 15. It indicates the potential of Oracle-MoE being a general-purpose solution.

We tested scenarios where the topic changes frequently. We randomly sample sentences from different datasets and combine them into a whole sequence. We observed that our proposed oracle space can still distinguish semantic groups efficiently, both in our models and public large MoE models, as shown in Figure 16 to Figure 19. We also tested the expert activation variation of such highly diverse data with Oracle-MoE and switch-transformer. On average, in every 100 consecutive token generations, Oracle-MoE only changes 12.20 times while the switch transformer changes

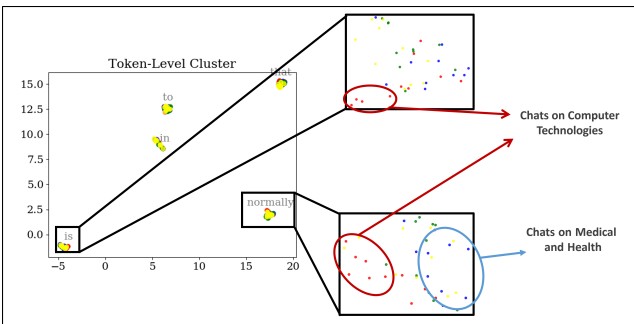

*Figure 10.* UMAP visualization of embedding space in DeepSeekMoE-16B from layer 10 on Wizard-of-Wikipedia datasets. Left: Tokens tend to cluster according to token-identity semantics. Right: Tokens from the same sequence are colored the same. They share similar semantics and stay closer to each other in each token cluster.

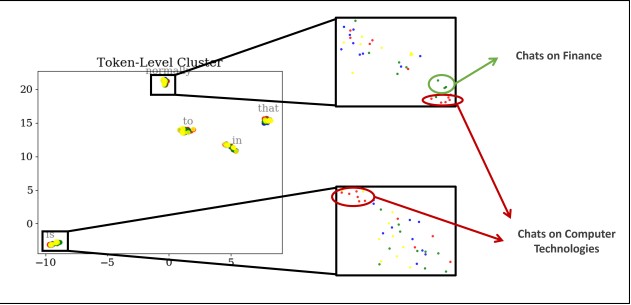

*Figure 11.* UMAP visualization of embedding space in DeepSeekMoE-16B from layer 15 on Wizard-of-Wikipedia datasets. Left: Tokens tend to cluster according to token-identity semantics. Right: Tokens from the same sequence are colored the same. They share similar semantics and stay closer to each other in each token cluster.

90.54 times. This is because in human natural language, it takes at least dozens of tokens to express a complete meaning, so our method still benefits from such "abrupt" semantic locality.

### B.3. MoE with fine-grained experts

We train a model following the setting of DeepSeekMoE-16B but with fewer parameters(3B): 12 MoE layers with 64 routed experts each as baseline, where hidden size is set to 1536 and expert intermediate size is set to 1024. The top 6 experts are selected for each token. Our method still achieved a 75% latency reduction at 2.5GB memory. Meanwhile, our model maintains the performance of downstream tasks. On Trivia QA, our model achieves an F1 Score of 50.20, compared to the baseline of 50.75. On XSum, our model attains a ROUGE-1 score of 21.74, while the baseline score is 21.22.

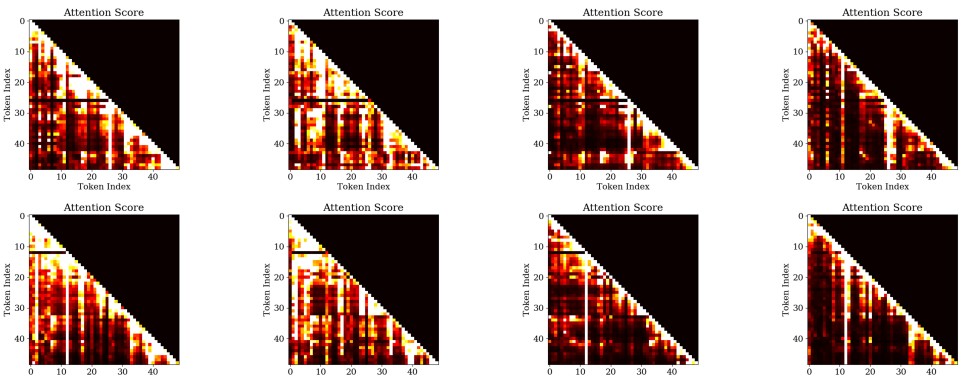

*Figure 12.* Attention scores on randomly sampled data of Wizard-of-Wikipedia(upper)and Synthetic-Persona-Chat(bottom) in DeepSeekMoE-16B from layers 5,10,15,20.

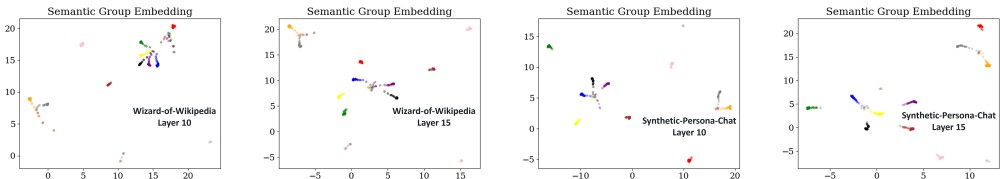

*Figure 13.* Semantic groups obtained by the Oracle-MoE method on Wizard-of-Wikipedia and Synthetic-Persona-Chat across different DeepSeekMoE-16B layers with semantic groups from the same sequence or user interaction are colored the same.

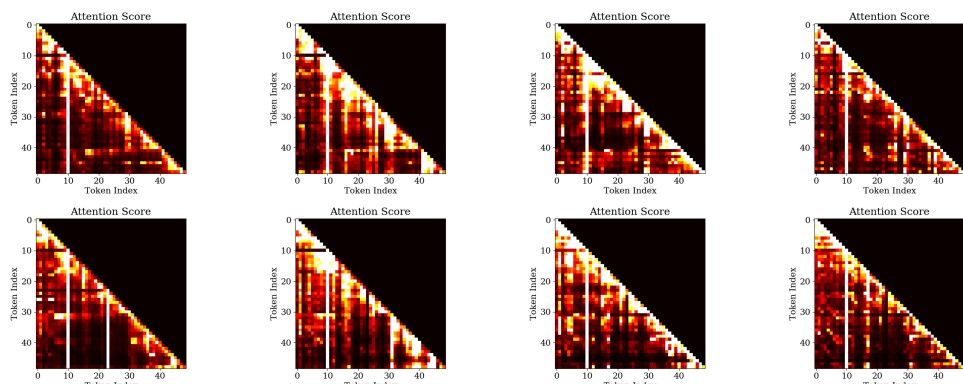

*Figure 14.* Attention scores on randomly sampled data of Wizard-of-Wikipedia(upper) and Synthetic-Persona-Chat(bottom) in Qwen1.5-MoE-A2.7B from layers 5,10,15,20.

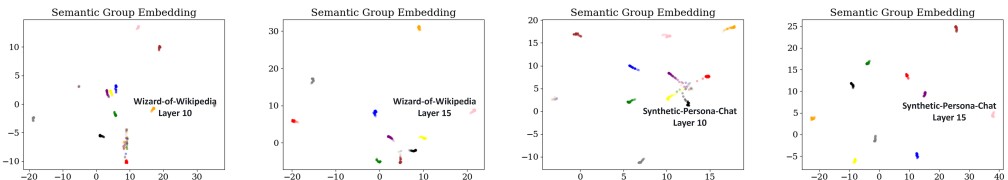

*Figure 15.* Semantic groups obtained by the Oracle-MoE method on Wizard-of-Wikipedia and Synthetic-Persona-Chat across different Qwen1.5-MoE-A2.7B layers with semantic groups from the same sequence or user interaction are colored the same.

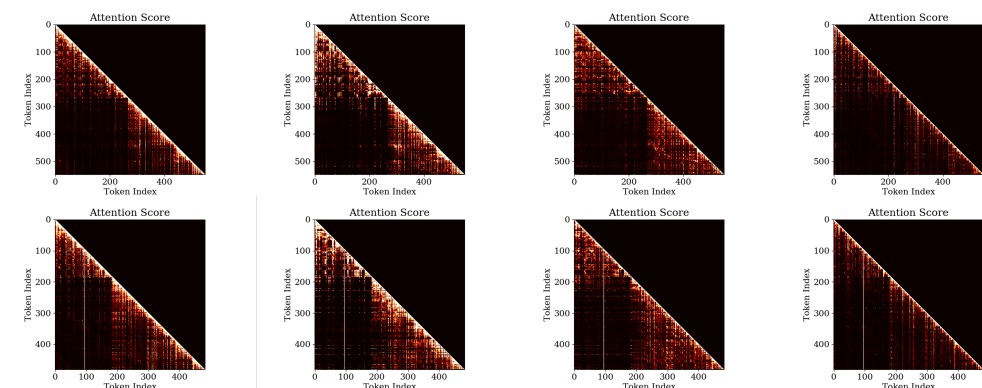

*Figure 16.* Attention scores on dierent diverse data(the top and bottom rows are different data) in DeepSeekMoE-16B from layers 5,10,15,20.

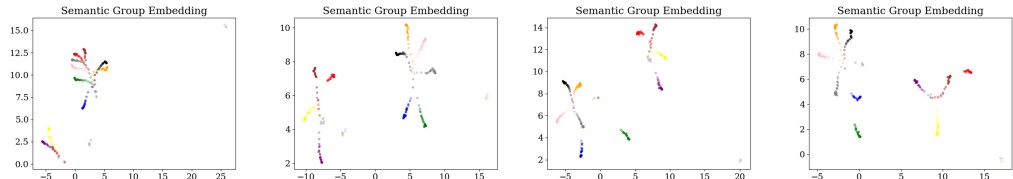

*Figure 17.* Semantic groups obtained by the Oracle-MoE method on diverse data from DeepSeekMoE-16B layers 5,10,15,20 with semantic groups from the same sequence or user interaction are colored the same.

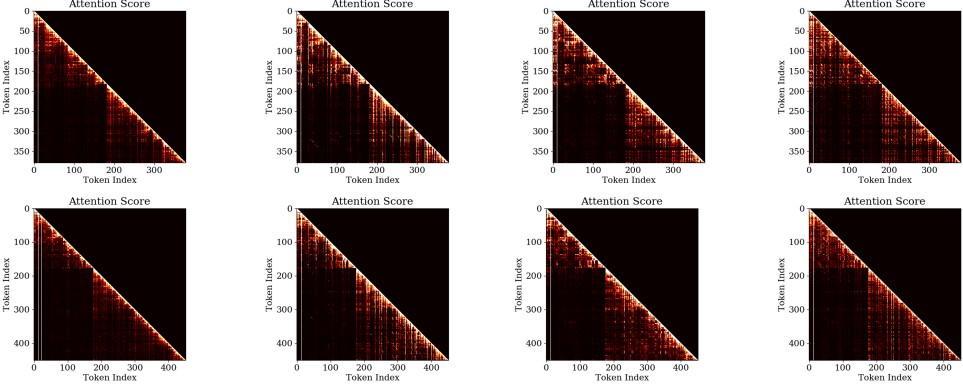

*Figure 18.* Attention scores on different diverse data(the top and bottom rows are different data) in Qwen1.5-MoE-A2.7B from layers 5,10,15,20.

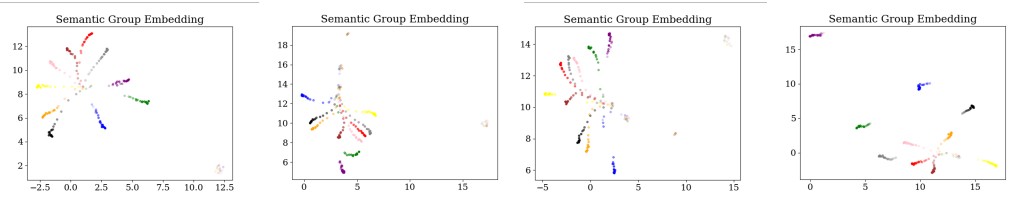

*Figure 19.* Semantic groups obtained by the Oracle-MoE method on diverse data from Qwen1.5-MoE-A2.7B layers 5,10,15,20 with semantic groups from the same sequence or user interaction are colored the same.

