# OpenReview forum: "Oracle-MoE: Locality-preserving Routing in the Oracle Space for Memory-constrained Large Language Model Inference"
_ICML.cc/2025/Conference — ICML 2025 poster_

### Official Review · Reviewer_bLcx · 2025-02-26

**Overall Recommendation:** 3

**Summary:**

This paper proposes a new MoE architecture, Oracle-MoE, to address the latency issues associated with deploying large language models (LLMs) on edge devices with limited memory.  The key idea is to route tokens in a compact space, called the oracle space, which is derived from attention scores to maintain semantic locality across tokens.  This approach reduces the frequent swapping of experts in and out of memory, which is a major cause of latency.  The paper provides theoretical analysis and experimental results on various GPT-2 models and downstream tasks, demonstrating that Oracle-MoE achieves state-of-the-art inference speeds without compromising task performance.
This work primarily targets the low throughput setting where "only a few activated experts <are required> to reside in memory for inference".

**Claims And Evidence:**

Claims with Clear and Convincing Evidence:

- High latency due to expert swapping: The paper provides clear evidence of the high latency caused by expert swapping in memory-constrained MoE-based LLM inference. They show that 50-85% of the latency is due to I/O overheads from expert swapping.
- Temporal inconsistencies in expert activations: The visualization of expert activations over a sequence of tokens clearly demonstrates the temporal inconsistencies, leading to frequent expert swapping.
- Oracle-MoE reduces expert activation variations: The paper provides evidence that Oracle-MoE effectively reduces expert activation variations compared to the Switch Transformer.
- Performance on downstream tasks: The results on various downstream tasks show that Oracle-MoE achieves similar or better performance compared to the Switch Transformer.

Claims that Need Further Support:

- Semantic locality: While the paper claims that tokens with higher mutual attention scores share similar high-level semantics, I am not convinced there is strong evidence of this.
- Effectiveness of oracle space: The paper claims that the oracle space efficiently describes various high-level semantics and that routing in this space preserves semantic locality; again, there is now strong evidence of this.


Overall, the paper presents a promising approach to address the latency challenges in memory-constrained MoE-based LLM inference. However, providing more detailed evidence and explanation for some of their claims would further strengthen their contribution. I'll also note that, if you're in the extremely memory-constrained MoE-based LLM inference setup, maybe an MoE is NOT the right solution.

**Essential References Not Discussed:**

not that i noticed

**Experimental Designs Or Analyses:**

The experimental designs and analyses in the paper are generally sound and valid. They provide convincing evidence for the effectiveness of Oracle-MoE in reducing latency without compromising performance. However, as mentioned earlier, providing more details on certain aspects, such as semantic locality and expert prediction, would further strengthen the paper's contributions.

**Methods And Evaluation Criteria:**

Yes, the proposed methods and evaluation criteria in the paper generally make sense for the problem of memory-constrained LLM inference on edge devices.

They define an Oracle Space motivated by Semantic Grouping of tokens. They cluster tokens in to groups in the oracle space; and they use Expert Prediction to see how much they can accelerate async loading of experts.

The chosen evals (Trivia QA, GLUE, MAG, Sci-Cite, XSum) are standard and relevant for evaluating the performance of LLMs on various downstream tasks; other evals could have been chosen, but these are ok. Some of the chosen evals are not the most standard in literature; hard to say what is / is not cherry picked.
Memory-Latency Curve and First Token Latency are used to measure the trade-off between memory usage and inference latency; these are reasonable choices.

The proposed methods and evaluation are good

**Other Comments Or Suggestions:**

none

**Other Strengths And Weaknesses:**

Narrow scope of application
If you're in a memory-constrained LLM inference setup, maybe you shouldn't be using MoEs.

**Questions For Authors:**

none

**Relation To Broader Scientific Literature:**

There are a lot of works (eg Mixtral) which look at expert assignment based on domains. These are not exactly the same as semantic meaning of tokens, but it should be mentioned and the difference explicitly noted from what you're doing.

**Theoretical Claims:**

I read theorem 1 and the skimmed the appendix proof. I have no strong statements about it.

---

> ### Author Rebuttal · Authors · 2025-03-31
>
> Thank you for your professional review comments and suggestions. We will add them in the updated version. Figures mentioned can be accessed at: https://anonymous.4open.science/r/ICML2025REBUTTAL-E158/README.md
>
> _Q1_ Semantic locality: While the paper claims that tokens with higher mutual attention scores share similar high-level semantics, I am not convinced there is strong evidence of this.
>
> _A1_ Previous studies [A][B] on representation space analysis have shown that semantically similar samples exhibit higher similarity in their embeddings compared to semantically dissimilar ones, which is also widely validated in experiments with general-purpose large models. We corroborate this observation and further identify a more fine-grained similarity pattern: token representations encapsulate both high-level semantics and token identity semantics. Among tokens with the same identity, the embeddings of those that share the same high-level semantic meaning tend to be more similar. This pattern is consistently observed in various models, including widely used large models like DeepSeek-16B-2.8B and Qwen1.5-MoE-A2.7B, which are illustrated in Figure 2 in our paper and Figure a in the link.
> Theoretical insights into how attention mechanisms compute correlations between tokens using the inner product of query (Q) and key (K) vectors are also supported by existing studies[C][D][E][F]. The computation of attention scores involves first assessing token correlations through inner products of query (Q) and key (K) vectors, followed by normalization of these correlations via softmax, and finally allocating contextual information through value (V) vectors weighted by the normalized scores. Among which, the Q-K inner product effectively captures token similarity and reflects high-level semantic alignment, as visualized in Figure b,d in the link.
>
> [A] A Survey on Word Embeddings: From Shallow Models to Deep Learning (Goldberg, 2017)
>
> [B] Deep Learning for NLP and Speech Recognition" (Hinton et al., 2012)
>
> [C] Exploring the Limits of Transfer Learning with a Unified Text-to-Text Transformer (Raffel et al., 2020)
>
> [D] Effective Approaches to Attention-based Neural Machine Translation(Luong et al., 2015)
>
> [E]Analyzing the Structure of Attention in Transformers (Kobayashi et al., 2020)
>
> [F]On the Expressive Power of Self-Attention Matrices （Valerii Likhosherstov et al.,2021）
>
> _Q2_ Effectiveness of oracle space: The paper claims that the oracle space efficiently describes various high-level semantics and that routing in this space preserves semantic locality; again, there is now strong evidence of this.
>
> _A2_ As mentioned above, Self-Attention, especially attention score, can reflect and amplify the high-level semantics across tokens in the same context. Therefore, oracle space, which is constructed by grouping tokens with high attention scores and extracting high-level semantics from them, can capture semantic locality well.
> To further demonstrate the generalizability of this conclusion, we constructed oracle space with DeepSeekMoE-16B\Qwen1.5-MoE-A2.7B on several datasets, as illustrated in Figure b-e in the link. We found that semantic group embeddings in these oracle spaces all preserve semantic locality well, with semantic group embeddings varying slowly and smoothly, showing the potential for scaling up and generalization.
>
> _Q3_ Narrow scope of application If you're in a memory-constrained LLM inference setup, maybe you shouldn't be using MoEs.
>
> _A3_ Pursuing the trade-off between performance and latency is a key topic in the field of LLM inference. The MoE structure, which scales up model performance without increasing the number of activated parameters, is the most edge-friendly model architecture. Recently, sparse MoE models like DeepSeek and QwQ have exhibited impressive ability, demonstrating the value of MoE edge deployment. Companies like Qualcomm have also started research concerning MOE edge deployment, which was presented at the NeurIPS 2024 enterprise section, illustrating the huge market value of this problem. It is believed that this technology will further increase the accessibility to LLMs at edge devices, better highlighting the advantages of MoE on edge devices.

---

> > ### Comment · Reviewer_bLcx · 2025-04-08
> >
> > Thank you for the response. Overall, the rebuttal hasn't shifted my views uch. The additional evidence and clarifications provided by the authors make a stronger case for the validity and potential impact of Oracle-MoE, but I'm going to keep my current rating.

---

### Official Review · Reviewer_fF5P · 2025-03-06

**Overall Recommendation:** 3

**Summary:**

This paper proposes Oracle-MoE, which improves the MoE inference efficiency by exploiting semantic locality to reduce swapping demands.

**Claims And Evidence:**

Please see **Other Strengths And Weaknesses**

**Essential References Not Discussed:**

Please see **Other Strengths And Weaknesses**

**Experimental Designs Or Analyses:**

Please see **Other Strengths And Weaknesses**

**Methods And Evaluation Criteria:**

Please see **Other Strengths And Weaknesses**

**Other Comments Or Suggestions:**

Please see **Other Strengths And Weaknesses**

**Other Strengths And Weaknesses:**

**Strengths**:

1. The paper is easy to follow, with clear writing and presentation.
2. The authors provide detailed theoretical analysis to justify their method.
3. The evaluation results are good and comprehensive.

**Weaknesses**:

1. The main concern I have with this paper is the issues of scalability. In the paper, the largest MoE models are 2B with 32 experts. However, the existing models like Mistral-8x7B [1] and DeepSeekMoE-16B [2] are much larger. It would be better if the authors could provide results on these models to ensure the scale-up and scale-out ability of the proposed method.

2. How would the number of sampled data in oracle space initialization impact the accuracy performance? Would this cause a potentially high overhead for larger models?

3. In terms of figure location, I suggest putting all figures on the top of the page instead of in between the texts (see Figure 2/3/4).

[1] Mixtral of Experts, Arxiv 2024.

[2] DeepSeekMoE: Towards Ultimate Expert Specialization in Mixture-of-Experts Language Models, Arxiv 2024.

**Questions For Authors:**

Please see **Other Strengths And Weaknesses**

**Relation To Broader Scientific Literature:**

Please see **Other Strengths And Weaknesses**

**Theoretical Claims:**

Please see **Other Strengths And Weaknesses**

---

> ### Author Rebuttal · Authors · 2025-03-31
>
> Thank you for your professional review comments and suggestions. We will add them in the updated version. Figures mentioned can be accessed at: https://anonymous.4open.science/r/ICML2025REBUTTAL-E158/README.md
>
> _Q1_ The main concern I have with this paper is the issue of scalability. In the paper, the largest MoE model is 2B with 32 experts. However, the existing models like Mistral-8x7B [1] and DeepSeekMoE-16B [2] are much larger. It would be better if the authors could provide results on these models to ensure the scale-up and scale-out ability of the proposed method.
>
> _A1_ Considering limited training resources and time, we train a model following the setting of DeepSeekMoE-16B but with fewer parameters(3B): 12 MoE layers with 64 routed experts each, where hidden size is set to 1536 and expert intermediate size is set to 1024. Top 6 experts are selected for each token. Our method still achieved a 75% latency reduction at 2.5GB memory. Meanwhile, our model maintains the performance of downstream tasks. On Trivia QA, our model achieves an F1 Score of 50.20, compared to the baseline of 50.75. On XSum, our model attains a ROUGE-1 score of 21.74, while the baseline score is 21.22.
> |Inference Latency|2.5GB|4.5GB|7GB|10GB|12GB(full)|
> |---|---|---|---|---|---|
> |Our Oracle Routing(s)|24.937|19.701|17.134|16.726|15.601|
> |Normal MoE(s)|113.828|89.082|44.013|27.058|15.601|
>
> _Q2_ How would the number of sampled data in oracle space initialization impact the accuracy performance? Would this cause a potentially high overhead for larger models?
>
> _A2_ As for the sampling overhead, the largest batch size we sampled in our experiments was 16382, and it only took 20 minutes for our 8*GTX3090 platform to process the sampling. Compared with 32.94 hours of training time, the sampling and oracle space construction contribute to only 1% of the overall wall-clock time, which is negligible. At the inference stage, our method introduces no inference overhead, thus there is no need to worry about larger models.
>
> _Q3_ In terms of figure location, I suggest putting all figures on the top of the page instead of in between the texts (see Figure 2/3/4).
>
> _A3_ Thank you for your suggestions. We will make such modifications to the layout of these pictures in the updated versions.

---

> > ### Comment · Reviewer_fF5P · 2025-04-02
> >
> > Thank you for your response. I will maintain my score.

---

### Official Review · Reviewer_jf9n · 2025-03-14

**Overall Recommendation:** 4

**Summary:**

This paper presents Oracle-MoE, a novel Mixture-of-Experts (MoE) architecture aimed at efficiently deploying Large Language Models (LLMs) on memory-constrained edge devices. Current MoE models, despite theoretical advantages for memory efficiency, suffer from high latency during inference due to frequent swapping of experts in and out of limited device memory. The authors identify that these latencies stem primarily from temporal inconsistencies in expert activations between consecutive tokens. To address this, Oracle-MoE introduces a new routing mechanism that groups tokens based on semantic similarity, routed within a compact "oracle space" defined by semantic group embeddings. This method significantly reduces the variability of expert activations, thereby minimizing expert swapping overhead and improving inference speed on devices with limited memory.

**Claims And Evidence:**

The Claims And Evidence are clear and well supported.

**Essential References Not Discussed:**

N/A

**Experimental Designs Or Analyses:**

The Experimental Designs Or Analyses are sound.

**Methods And Evaluation Criteria:**

The paper proposes routing based on semantic embeddings derived from attention score clustering. In general it's innovative and well justified. However Oracle-MoE initialization requires a warm-up stage and clustering analysis (e.g., via K-means and SVD), which introduces complexity. Evaluation is done primarily on the NVIDIA Jetson Xavier NX, which could limit the scope of edge devices. Broader evaluations could enhance confidence in the method’s universality.

**Other Comments Or Suggestions:**

"Lanuge" -> Language in title

**Other Strengths And Weaknesses:**

Strengths
- The paper clearly identifies a practical limitation—high latency from expert swapping—providing insightful analysis into its underlying cause (temporal inconsistency).
- The proposal of routing tokens based on semantic locality rather than token-specific embeddings is innovative and well-motivated
- The authors offer a robust theoretical framework around Consecutive Semantic Difference (CSD)
- A detailed breakdown of latency composition provides clear empirical insights into exactly how Oracle-MoE improves performance over baseline strategies.
- Oracle-MoE is novel in its introduction of an oracle-space routing approach, leveraging semantic locality in attention scores

Weaknesses
- The paper could benefit from further discussions on practical deployment considerations and real-world constraints, such as more diverse hardware scenarios beyond the NVIDIA Jetson platform, like A100s and H100s.
- The concept of predicting expert activations in deeper layers based on shallow layers is promising, but the rationale and robustness of the 85%-95% prediction accuracy are left for future exploration. More details on this approach would strengthen the work significantly.
- While semantic locality is effectively leveraged, the paper does not deeply investigate scenarios where semantic locality is minimal (highly diverse or abrupt topic changes), potentially limiting generalizability.
- The paper did not discuss cases of using fine-grained experts ( e.g., number of experts > 128, like the ones in DeepSeek's model).

**Questions For Authors:**

N/A

**Relation To Broader Scientific Literature:**

This is going to have most impact to the budget-constrained inference domain.

**Theoretical Claims:**

The Theoretical Claims are well justified.

---

> ### Author Rebuttal · Authors · 2025-03-31
>
> Thank you for your professional review comments and suggestions. We will add them to the updated version. The figures mentioned can be accessed at: https://anonymous.4open.science/r/ICML2025REBUTTAL-E158/README.md
>
> _Q1_ The paper could benefit from further discussions on practical deployment considerations and real-world constraints, such as more diverse hardware scenarios beyond the NVIDIA Jetson platform, like A100s and H100s.
>
> _A1_ Results on A100s are listed below. Our method still speeds up inference by 50%~350%.
> |Model size|Memory budget|Switch(FIFO)|Switch(LRU)|Switch(SwapMoE)|Ours(FIFO)
> |---|---|---|---|---|---|
> |9*24(2.06B)|1GB|16.613s|16.066s|15.900s|4.012s|
> ||2GB|13.762s|15.229s|12.751s|3.630s||
> ||4GB|11.165s|11.973s|9.754s|3.529s||
> ||7GB|6.688s|7.115s|6.135s|3.410s||
> ||Full Memory|3.252s|3.252s|3.252s|3.252s||
>
> _Q2_ The concept of predicting expert activations in deeper layers based on shallow layers is promising, but the rationale and robustness of the 85%-95% prediction accuracy are left for future exploration. More details on this approach would strengthen the work significantly.
>
> _A2_ We predict by training a linear classifier with representations in shallow layers as input and routing results in deep layers as target labels. It turns out that such a linear classifier can reach an accuracy of 85%\~95% in our structure and 40%\~60% in existing MoE structures. More fine-grained observations show that layers that are closer predict each other well, e.g., representations in layer 10 predict routing results in layer 12 better than that of layer 5, and layers that predict each other well show a grouped pattern. We have been studying this phenomenon and believe this can be attributed to residual connections maintaining semantics across layers to some extent.
> |Avg Prediction Acc from|Qwen|Switch(Ourtrained)|Oracle(Ourtrained)|
> |---|---|---|---|
> |Layer 0|48.23|50.98|89.66|
> |Half of the model layers|55.59|59.72|92.82|
> |3/4 of the model layers|63.27|68.68|95.51|
>
> _Q3_ While semantic locality is effectively leveraged, the paper does not deeply investigate scenarios where semantic locality is minimal (highly diverse or abrupt topic changes), potentially limiting generalizability.
>
> _A3_ We tested scenarios where the topic changes frequently. We randomly sample sentences from different datasets and combine them into a whole sequence. We observed that our proposed oracle space can still distinguish semantic groups efficiently, both in our models and public large MOE models (DeepSeek-16B-2.8B\Qwen1.5-MoE-A2.7B), as shown in Figure g,h,i,j in the link. We also tested the expert activation variation of such highly diverse data with Oracle-MoE and switch-transformer. On average, in every 100 consecutive token generations, Oracle-MoE only changes 12.20 times while the switch transformer changes 90.54 times. This is because in human natural language, it takes at least dozens of tokens to express a complete meaning so our method still benefits from such "abrupt" semantic locality.
>
> _Q4_ The paper did not discuss cases of using fine-grained experts ( e.g., number of experts > 128, like the ones in DeepSeek's model).
>
> _A4_ Considering limited training resources and time, we train a model following the setting of DeepSeekMoE-16B but with fewer parameters(3B): 12 MoE layers with 64 routed experts each, where hidden size is set to 1536 and expert intermediate size is set to 1024. The top 6 experts are selected for each token. Our method still achieved a 75% latency reduction at 2.5GB memory. Meanwhile, our model maintains the performance of downstream tasks. On Trivia QA, our model achieves an F1 Score of 50.20, compared to the baseline of 50.75. On XSum, our model attains a ROUGE-1 score of 21.74, while the baseline score is 21.22.
> |Inference Latency|2.5GB|4.5GB|7GB|10GB|12GB(full)|
> |---|---|---|---|---|---|
> |Our Oracle Routing(s)|24.937|19.701|17.134|16.726|15.601|
> |Normal MoE(s)|113.828|89.082|44.013|27.058|15.601|
>
> _Q5_ typo "Lanuge".
>
> _A5_ We will correct the typos in the updated version.

---

### Official Review · Reviewer_dPcM · 2025-03-17

**Overall Recommendation:** 3

**Summary:**

The paper introduces Oracle MoE, a novel Mixture-of-Experts (MoE) architecture designed specifically for memory-constrained inference on edge devices. The main idea is to replace conventional token level routing with an oracle-space routing mechanism that leverages semantic locality. By grouping tokens based on their high level semantic similarity (extracted via attention scores) and using compact semantic group embeddings (the “oracle space”), the method reduces the frequency of expert swapping and thus significantly lowers inference latency. Extensive experiments on GPT 2 based models across various sizes and downstream tasks show that Oracle MoE achieves state of the art latency improvements while maintaining competitive task performance.

**Claims And Evidence:**

Some claims are supported by clear and convincing evidence. However, there are some important claims are problematic. For example,

(1) the temporal inconsistencies of inter-token expert activation are not clear. The authors only show visualization results for a specific sample. It would be better to measure the temporal inconsistencies for the whole dataset and different layers.

(2) The key assumption for semantic locality, i.e., linguistic meaning between consecutively generated tokens is typically consistent, is not supported by any evidence.

(3) Can the authors explain more about why the mapping of Q/K/attention score will group consecutive tokens with similar semantics? Why does this happen for different layers/samples?

**Essential References Not Discussed:**

Yes.

**Experimental Designs Or Analyses:**

Yes. I check the experimental design and analysis. Although the authors provide memory-latency results and demonstrate the overall performance for QA, classification and summarization tasks, there are several important experiment are missing:

(1)	The experimental result on the temporal inconsistency of inter-token expert activation is missing. It would be more convincing to provide a metric to quantitatively measure the temporal inconsistency and show how Oracle-MOE reduces this inconsistency.

(2)	The semantic locality, as a key assumption, is not well explained. It would be better to provide some preliminary experiments to show the evidence in real datasets. It would be interesting to see the semantic locality across different models/layers/samples.

(3)	There are some approximations in the derivation of oracle-MOE, such as CSD_token in Line 147, page 3, CSD_oracle in Line 231, page 5. It would be better to validate such approximation in the experiments design.

(4)	Ablation study on CSD and hyperparameter study of $\gamma$ are missing. How does $\gamma$ influence the proposed algorithm?

**Methods And Evaluation Criteria:**

The proposed method makes sense to reduce latency via eliminating massive swapping demands.

**Other Comments Or Suggestions:**

N/A

**Other Strengths And Weaknesses:**

Strengths:

1.	The key idea of maintaining the semantic locality across consecutive tokens is very simple and efficient in terms of latency.

2.	This paper is well-organized and easy to read. The paper first shows the latency bottleneck on expert swapping for memory-constrained scenario, and then formulates latency optimization problem with a simple yet effective solution. The overall organization is easy to follow and smooth.

3.	The authors demonstrate the effectiveness of the proposed method across different tasks and provide some visualization results (such as attention, semantic group embedding) to support some claims.

Weakness:

(1)	The experimental result on the temporal inconsistency of inter-token expert activation is missing. It would be more convincing to provide a metric to quantitatively measure the temporal inconsistency and show how Oracle-MOE reduces this inconsistency.

(2)	The semantic locality, as a key assumption, is not well explained. It would be better to provide some preliminary experiments to show the evidence in real datasets. It would be interesting to see the semantic locality across different models/layers/samples.

(3)	There are some approximations in the derivation of oracle-MOE, such as CSD_token in Line 147, page 3, CSD_oracle in Line 231, page 5. It would be better to validate such approximation in the experiments design.

(4)	Ablation study on CSD and hyperparameter study of $\gamma$ is missing. How does $\gamma$ influence the proposed algorithm?

**Questions For Authors:**

N/A

**Relation To Broader Scientific Literature:**

The key contributions can accelerate the MOE model inference in edge devices scenario.

**Theoretical Claims:**

I don’t check the theoretical claims carefully.

---

> ### Author Rebuttal · Authors · 2025-03-31
>
> Thank you for your professional review comments and suggestions. We will add them in the updated version. Figures mentioned can be accessed at: https://anonymous.4open.science/r/ICML2025REBUTTAL-E158/README.md
>
> _Q1_ It would be better to measure the temporal inconsistencies for the whole dataset and different layers. It would be more convincing to provide a metric to quantitatively measure the temporal inconsistency and show how Oracle-MOE reduces this inconsistency.
>
> _A1_ We propose temporal activation inconsistency, defined as the average number of inconsistent expert activations per 100 consecutive tokens per expert. Results over the entire dataset and across different models and layers are listed below. Existing MoEs show strong temporal activation inconsistency within all layers, while Oracle-MoE reduces this.
>
> |Activation inconsistency|DeepSeek|Qwen|Switch|Oracle|
> |--|--|--|--|--|
> |1st 1/4 layers avg|80.84|81.56|69.20|6.03|
> |2nd 1/4 layers avg|65.35|71.04|64.87|4.82|
> |3rd 1/4 layers avg|70.68|75.37|53.36|4.20|
> |4th 1/4 layers avg|76.61|77.16|75.44|5.11|
>
> _Q2_ It would be better to provide some preliminary experiments to show the evidence of semantic locality in real datasets, .. across different models/layers/samples.
>
> Experiments with DeepSeekMoE-16B and Qwen1.5-MoE-A2.7B on real chat datasets(Wizard-of-Wikipedia and Synthetic-Persona-Chat) are shown in Figure a-e in the link. Semantic locality appears across different models/layers/samples. Semantic groups can still be distinguished based on attention score and obtained by our method. It indicates the potential of Oracle-MoE being a general-purpose solution.
>
> _Q3_ Can the authors explain more about why the mapping of Q/K/attention score will group consecutive tokens with similar semantics? Why does this happen for different layers/samples?
>
> _A3_ Previous studies [A][B] have shown that semantically similar samples exhibit higher embedding similarity than semantically dissimilar ones, widely validated in experiments with large models. We corroborate this observation and identify a more fine-grained pattern: token representations encapsulate high-level and token identity semantics. Among tokens with the same identity, embeddings of those sharing the same high-level semantics tend to be more similar. This pattern is consistently observed in models like DeepSeek-16B-2.8B and Qwen1.5-MoE-A2.7B, as illustrated in Figure 2 in the paper and Figure a in the link.
> Theoretical insights are also supported by [C][D]. Computing attention scores involves first assessing token correlations through inner products of Q and K vectors, normalization via softmax, and allocating contextual information through V weighted by the normalized scores. Among these, the Q-K inner product effectively captures token similarity and reflects high-level semantic alignment, as visualized in Figure b,d in the link.
>
> [A] A Survey on Word Embeddings: From Shallow Models to Deep Learning (Goldberg, 2017)
>
> [B] Deep Learning for NLP and Speech Recognition (Hinton et al., 2012)
>
> [C] Exploring the Limits of Transfer Learning with a Unified Text-to-Text Transformer (Raffel et al., 2020)
>
> [D] Analyzing the Structure of Attention in Transformers (Kobayashi et al., 2020)
>
> _Q4_ There are some approximations in the derivation of oracle-MOE. It would be better to validate such an approximation in the experimental design.
>
> _A4_ Experiments validating this approximation are listed below: for token pairs with the distance between [x,y), we count on average how many inconsistent expert activations are triggered. There is a statistically positive correlation that the higher the distance between tokens, the more likely they activate different experts. More results are in Figure f in the link.
>
> |Embedding L2 Distance|[0.0,22)|[22,33)|[33,44)|[44,55)|[55,66)|[66,77)|[77,88)|[88,+∞)|
> |---|---|---|---|---|---|---|---|---|
> |Inconsistent activation per token|0.0|0.083|0.222|0.377|0.576|0.770|0.878|0.903|
>
> _Q5_ Ablation study on CSD and hyperparameter study of γ are missing. How does γ influence the proposed algorithm?
>
> _A5_ CSD and γ are key metrics we defined for the optimization problem of minimizing the model's consecutive inconsistent activation while maintaining its task performance above γ. In our paper, we design oracle space routing to reduce CSD while maintaining γ. We found that the number of MoE layers and experts per layer can influence CSD and γ as is listed in the table below. CSD is given as the average number of inconsistent expert activations per 100 tokens in all layers and γ is given as the average of all tasks tested. Our model has a larger γ and less CSD,  performing consistently well at large expert numbers, demonstrating our method's robustness to hyperparameters.
>
> |Model Scale|2*4(192M)|4*8(295M)|8*16(729M)|9*24(2.06B)|
> |--|--|--|--|--|
> |our model's outperform baseline's γ by(larger better)|+0.02|+0.60|+0.49|+0.36|
> |our model reduces CSD by(smaller better)|-25.10|-53.94|-71.23|-85.41|

---

> > ### Comment · Reviewer_dPcM · 2025-04-07
> >
> > Thanks for the detailed and thoughtful response. I have increased my score to 3 accordingly.

---

### Decision · Program_Chairs · 2025-05-01

**Decision:**

Accept (poster)

**Comment:**

This paper presents Oracle-MoE, a memory-efficient Mixture-of-Experts architecture that exploits semantic locality to reduce inference latency. All four reviewers expressed positive opinions, i.e., three with Weak Accept and one with Accept and the rebuttal addressed key concerns, including semantic locality, temporal inconsistency, and scalability, thus prompting one reviewer to raise his/her score. Given the overall consensus and the paper’s well-structured contributions, I would recommend acceptance.